# *N*-Acetylcysteine-Loaded Magnetic Nanoparticles for Magnetic Resonance Imaging

**DOI:** 10.3390/ijms241411414

**Published:** 2023-07-13

**Authors:** Martina Kubovcikova, Radka Sobotova, Vlasta Zavisova, Iryna Antal, Iryna Khmara, Maksym Lisnichuk, Zuzana Bednarikova, Alena Jurikova, Oliver Strbak, Jana Vojtova, Pavol Mikolka, Jan Gombos, Alica Lokajova, Zuzana Gazova, Martina Koneracka

**Affiliations:** 1Institute of Experimental Physics, Slovak Academy of Sciences, Watsonova 47, 04001 Kosice, Slovakia; kubovcikova@saske.sk (M.K.); sobotova@saske.sk (R.S.); zavisova@saske.sk (V.Z.); iryna.antal@saske.sk (I.A.); khmara@saske.sk (I.K.); bednarikova@saske.sk (Z.B.); akasard@saske.sk (A.J.); gazova@saske.sk (Z.G.); 2Faculty of Science, Pavol Jozef Safarik University, Park Angelinum 9, 04001 Kosice, Slovakia; maksym.lisnichuk@upjs.sk; 3Jessenius Faculty of Medicine in Martin, Comenius University in Bratislava, Mala Hora 4, 03601 Martin, Slovakia; oliver.strbak@centrum.cz (O.S.); jana.vojtova@uniba.sk (J.V.); pavol.mikolka@uniba.sk (P.M.); gombos9@uniba.sk (J.G.);

**Keywords:** magnetic nanoparticles, *N*-acetylcysteine adsorption, physicochemical characterization, MRI, relaxometry, relaxation time, relaxivity

## Abstract

Acute respiratory distress syndrome (ARDS) is a life-threatening condition characterized by the rapid onset of lung inflammation Therefore, monitoring the spatial distribution of the drug directly administered to heterogeneously damaged lungs is desirable. In this work, we focus on optimizing the drug *N*-acetylcysteine (NAC) adsorption on poly-l-lysine-modified magnetic nanoparticles (PLLMNPs) to monitor the drug spatial distribution in the lungs using magnetic resonance imaging (MRI) techniques. The physicochemical characterizations of the samples were conducted in terms of morphology, particle size distributions, surface charge, and magnetic properties followed by the thermogravimetric quantification of NAC coating and cytotoxicity experiments. The sample with the theoretical NAC loading concentration of 0.25 mg/mL was selected as an optimum due to the hydrodynamic nanoparticle size of 154 nm, the surface charge of +32 mV, good stability, and no cytotoxicity. Finally, MRI relaxometry confirmed the suitability of the sample to study the spatial distribution of the drug in vivo using MRI protocols. We showed the prevailing transverse relaxation with high transverse relaxivity values and a high *r*_2_^(^*^)^/*r*_1_ ratio, causing visible hypointensity in the final MRI signal. Furthermore, NAC adsorption significantly affects the relaxation properties of PLLMNPs, which can help monitor drug release in vitro/in vivo.

## 1. Introduction

Aspiration-induced lung injury is often underdiagnosed in the clinical setting in critically ill patients. It is considered an independent risk factor for the subsequent development of acute respiratory distress syndrome (ARDS) [1]. The era of COVID-19 reminds us how difficult it is to treat patients with this syndrome—patients with ARDS are provided with supportive ventilation to ensure sufficient body oxygenation. However, the biophysical forces acting on artificial lung ventilation can contribute to an increased inflammatory activity and alveolar–capillary membrane damage, a phenomenon known as ventilator-induced lung injury [2]. Lung damage can be affected by the local administration of the drug directly into the lungs, thereby avoiding the undesirable effects that may occur with a systemic therapeutic approach. Nevertheless, such direct drug application is relatively difficult because the respiratory system has defense mechanisms to retain inhaled drug particles and remove or inactivate them [3]. Therefore, the knowledge of the distribution of the drug applied directly to the lungs is more than desirable. Imaging the distribution of drugs in human tissue is one of the current problems being solved in biomedical research [4]. The basic but also the most frequently used method is magnetic resonance imaging (MRI) [5], which enables imaging at the molecular level [6]. This method is non-invasive and has minimal side effects due to the absence of ionizing radiation. Therefore, it can be used repeatedly, even in high-risk patients such as pregnant women or children. Due to the weak intensity of the MRI signal from the lung parenchyma, its use in clinical practice is very limited, so contrast and non-contrast perfusion imaging have been developed [7]. The use of aerosolized contrast agents may be a breakthrough in lung MRI [8]. On the one hand, they would provide an image of the pathology in the tissue; on the other hand, they would inform about the success of drug access to the desired site and the progress of the treatment [9]. However, this requires functionalizing specific contrast agents and optimizing MRI pulse sequences. The use of magnetic nanoparticles (MNPs) for this purpose is an idea that still arouses unceasing interest [10,11]. Recently, several studies have appeared, intending to use magnetic nanoparticles for the MRI of lung cancer [12,13,14]. However, they only encompass a review or pilot study that does not include a comprehensive methodology of the MRI of the lungs or the MRI of the spatial distribution of drugs in the lungs using aerosols. Therefore, our main MRI goal is to test MRI protocols for the volumetric and relaxometric imaging of the spatial distribution of drugs bound to MNPs in the lungs, which, to our knowledge, has not yet been published.

Magnetic nanoparticles are promising nanostructures for application in diagnosis and therapy due to their biocompatibility, simple fabrication technology, and the ability to be manipulated with an external magnetic field [15]. In addition to biocompatibility, MNPs used in medical applications must have other properties—small size, monodispersity, a fast magnetic response in a magnetic field, no residual magnetism, and good stability in biological media. The most frequently used magnetic material with a high magnetic moment value is magnetite. Magnetite can be prepared by several methods, but the most widely used one is the method whose principle rests on the precipitation of ferric and ferrous salts in an alkaline environment. This reaction results in the formation of magnetic nanoparticles (Fe_3_O_4_) with an average magnetic core size of about 10 nm. To ensure the stability of MNPs during in vivo use, MNPs have to be coated with a suitable substance or functionalized with suitable functional groups. Functionalization is not only a prevention of aggregation, but also a means of ensuring the biocompatibility of nanoparticles used in medicine. One of the molecules that can be used for the surface stabilization, modification, and coating of MNPs are the amino acids due to their nontoxic and biocompatible features. Furthermore, the presence of an amino acid on the surface of nanoparticles can ensure the biological identity of the nanoparticles; therefore, they are also known as smart nanomaterials [16] that could be used for numerous biological applications such as drug delivery, bioimaging, biosensing, etc. [17]. We used poly-l-lysine (PLL) in our functionalization process [18,19]. PLL is a homopolypeptide belonging to the group of cationic polymers at pH 7 that contains a positively charged hydrophilic amino group bound to the surface of MNPs through the carboxyl group to develop an electrostatic interaction.

*N*-Acetylcysteine (NAC) is a drug approved by the Food and Drug Administration (FDA) and recognized by the World Health Organization (WHO) as an essential drug widely used for the treatment of acetaminophen overdose (paracetamol) and, more recently, as a mucolytic in respiratory diseases as well. The primary role of NAC is associated with its antioxidant and anti-inflammatory activity, which favors the maintenance of a cellular redox imbalance. For this reason, its therapeutic potential concerns a series of diseases that link oxidative stress with its etiology and progression. However, the mechanisms by which NAC exerts its antioxidant and cytoprotective capacity under different physiological conditions have not been fully clarified. The growing interest in investigating the favorable effects of NAC involves not only its action as a potent cell bio-protector, but also its pharmacokinetic characteristics related to safety, absorption, and bioavailability associated with its low cost [20]. Furthermore, *N*-acetylcysteine is a mucolytic drug commonly used for respiratory tract treatment. The drug dissolves all the components that cause mucus to become viscous and thus promotes expectoration. The drug is also used to treat respiratory diseases accompanied by an intense formation of dense, viscous mucus. It also exhibits significant antiviral activity against influenza viruses [21]. NAC can be administered orally, intravenously, or by inhalation, which is commonly safe and well-tolerated, even in high doses [20].

In this paper, we focus on optimizing the drug *N*-acetylcysteine adsorption on amino-functionalized iron oxide magnetic nanoparticles (PLLMNPs) to prepare nanomaterial which will enable magnetic resonance imaging of the drug distribution in the lungs. The physicochemical characterizations of the prepared nanoparticles with and without adsorbed NAC were conducted in terms of morphology, particle size distribution, and surface charge, followed by the evaluation of NAC adsorption and time stability of the samples. The cytotoxicity of the samples was investigated using human endothelial kidney (HEK 293) cell lines by the Lactate dehydrogenase (LDH) assay kit at different particle concentrations for 24 and 48 h incubation time. Finally, MRI relaxometry confirmed the suitability of prepared NAC-PLLMNPs to study the spatial distribution of the drug in vivo by MRI protocols. We showed the prevailing transverse relaxation with high transverse relaxivity values and a high *r*_2_^(^*^)^/*r*_1_ ratio, causing visible hypointensity in the final MRI signal. Furthermore, NAC adsorption significantly affects the relaxation properties of PLLMNPs, which can help monitor drug release in vitro/in vivo.

## 2. Results

To study the NAC adsorption on the surface of PLLMNPs, a set of eight samples was prepared with different theoretical weight ratios of NAC/MNPs ranging from 1 to 20 *w*/*w*. Then, the effect of the theoretical NAC/MNPs weight ratio on size, zeta potential, and isoelectric point (IEP) was investigated by different methods.

### 2.1. Zeta Potential Measurements

The effect of various theoretical weight ratios of NAC/MNPs on zeta potential and IEP is shown in Figure 1A,B. Based on the analysis of these figures, the zeta potential slightly decreases after the addition of NAC until it reaches the point NAC/MNPs of ~5. Above this value, a sharp decrease in the zeta potential is observed, which leads to an assumption that the optimum theoretical NAC/MNPs weight ratio is close to 5 (Figure 1A). The shift of the zeta potential distributions with increasing NAC/MNPs weight ratio is shown in the inset of Figure 1A. The dependence of the IEP of the samples as a function of the theoretical weight ratios of NAC/MNPs also confirmed that the optimal ratio of the NAC bound to the PLLMNPs is close to 5 (Figure 1B). The IEP values were obtained by measuring each sample’s zeta potential as a function of pH and by identifying the pH where the zeta potential value crosses zero. The inset in Figure 1B shows the IEP of three selected NAC-loaded PLLMNPs samples (NAC-PLLMNPs-1, NAC-PLLMNPs-5, and NAC-PLLMNPs-15), NAC, and PLLMNPs alone. The IEP value of the drug-free PLLMNPs sample (~9.6) was shifted toward lower values with increasing drug concentration, indicating NAC adsorption onto PLLMNPs.

The observed shift in IEP (toward the IEP value of the pure NAC solution ~2.6 [22]) indicates its direct correlation with the degree of surface coverage of the nanoparticle with NAC. From these findings, the value of 5 seemed to be the optimal ratio of NAC to MNPs. However, the sample NAC-PLLMNPs-5 did not exhibit good stability. The reason for the behavior can result in an excess of added NAC, which leads to the formation of additional NAC layers and a less stable sample. Based on the outcome, NAC-PLLMNPs-1 and NAC-PLLMNPs-3 samples were chosen for the subsequent characterizations.

### 2.2. Morphology and Size Distribution

Transmission electron microscopy (TEM) is one of the most powerful analytical methods available to provide direct structural and morphological information about magnetic nanoparticles. Using an electron beam with a very short wavelength, it is possible to examine the structure of magnetic nanoparticles in detail down to the atomic level. Figure 2A,B,D,E shows the typical TEM images of prepared PLLMNPs and NAC-PLLMNPs-3 samples, where it is possible to see a quasi-spherical shape of the nanoparticles’ core with an average diameter of *D*_TEM_ = 10.4 ± 0.1 nm (standard deviation σ = 2.66; the number of particles counted, *n* = 884) for PLLMNPs and of 10.8 ± 0.2 nm (σ = 2.70; *n* = 933) for NAC-PLLMNPs-3 as determined by TEM (Figure 2C,F).

A thin layer of PLL and NAC coating can be seen in the pictures. As a result of sample drying, a necessary step in sample preparation for TEM analysis, the agglomeration of nanoparticles is also observed.

The size distributions of NAC-PLLMNPs-3 and bare PLLMNPs were also determined using atomic force microscopy (AFM) (Figure 3). The AFM image and the histogram of PLLMNPs size distribution (*D*_AFM_ = 11.6 ± 0.6 nm) show that the addition of PLL increases the particles’ size (Figure 3A) as the diameter is larger than the diameter of the magnetite core observed by TEM (10.4 ± 0.1 nm). The NAC-PLLMNPs-3 diameter is even higher, with 13.7 ± 2.1 nm compared to PLLMNPs confirming the NAC binding to the surface of the PLLMNPs (Figure 3B). However, all samples form clusters with variable size distributions.

The next technique currently used for the characterization of nanomaterials in terms of size distribution was dynamic light scattering (DLS). The particle size distributions of the chosen samples were measured in deionized water by DLS at 25 °C and the results are summarized in Table 1. As seen in the table, with the increasing theoretical NAC/MNPs *w*/*w* ratio, the Z-average particle size (*D*_DLS_) starts to increase from 112 nm for PLLMNPs to 154 nm and 238 nm for NAC-PLLMNPs-1 and NAC-PLLMNPs-3, respectively. This indicates that the change in the particle size occurs due to the successful NAC adsorption on the amino-modified magnetic nanoparticle surface.

A comparison of particle size obtained from DLS with direct microscopic measurements is highly complicated, and cannot be performed directly [23]. DLS is used to characterize the solutions from which the samples for TEM and AFM microscopy were made. This technique is very different from the imaging of dried samples and is sensitive to the dynamic aggregation, aggregation, and agglomeration behavior of the particles in solution [24,25]. Furthermore, the measurement of sizes from DLS data is an indirect method, based on the determination of the frequency of movement and the modeling of the size from these data. Moreover, DLS, unlike TEM, provides us with information about the hydrodynamic radii of the particles, which include not merely the particle itself, but the ionic and solvent layers associated with it in solution. Therefore, it can be summarized that the relatively large differences in size obtained by DLS and microscopic techniques may be attributed to the abovementioned facts.

### 2.3. Magnetic Measurements

Superparamagnetism, that is the responsiveness to an applied magnetic field without permanent magnetization, is an essential property for applying MNPs. Therefore, these magnetic properties are critical in the magnetic nanoparticle applications of the biomedical and bioengineering fields. The magnetic characterizations of the naked MNPs, PLLMNPs, and NAC-adsorbed PLLMNPs samples are shown in Figure 4A with typical characteristics of superparamagnetic behavior, i.e., no coercivity and remanence were detectable at room temperature. From Figure 4A, the saturation magnetization (*M*_S_) of the MNPs sample decreases from 73.6 emu/g to 65.0, 64.4 and 63.4 emu/g for PLLMNPs, NAC-PLLMNPs-1, and NAC-PLLMNPs-3 samples, respectively. This reduction in *M*_S_ in coated samples is attributed to the non-magnetic layer on the MNPs surface. This is because magnetization is proportional to the amount of weight for the same magnetic material. Increasing the coating layer increases the amount of nonmagnetic material on the MNPs. It means that the more the layers of coating, the less the amount of magnetite contained in the same weight unit of the sample. In addition, by dividing the *M*_s_ of the NAC-loaded samples by the *M*_s_ of the MNPs (73.6 emu/g), we can estimate that the magnetite content of NAC-PLLMNPs-1 and NAC-PLLMNPs-3 samples is 99 wt% and 97 wt%, respectively. These outcomes can serve to calculate the mg NAC adsorbed on mg MNPs surface (see Table 1).

The magnetization curve allows us to determine the magnetic core diameter (*D*_MAG_) by fitting to the Langevin function as well [26]. Figure 4B displays the magnetization curves fitted by the Langevin function (red line) and corresponding magnetic core particle size distribution with mean *D*_MAG_. The calculated *D*_MAG_ is almost the same for all samples (~10 nm) leading us to conclude that PLL functionalization and NAC adsorption does not impair the magnetic behavior of the MNPs.

### 2.4. Thermogravimetric Analysis (TGA)

The decomposition behavior of the bare MNPs, PLLMNPs, and NAC-loaded PLLMNPs samples with two input NAC/MNPs *w*/*w* ratios (NAC-PLLMNPs-1, NAC-PLLMNPs-3) is shown in Figure 5. As seen from the TG (Figure 5A) and dTG (Figure 5B) curves, only a tiny mass loss was observed up to 100 °C for all investigated samples due to initial water evaporation. Then, the main decomposition stage is followed up to about 450 °C, depending on the sample. Above this temperature, no significant mass loss was observed. The percentage mass loss in the TGA curves at a temperature of 700 °C found that approximately 0.8 and 3.3% NAC are bound to the PLLMNPs for the NAC-PLLMNPs-1 and NAC-PLLMNPs-3 samples, respectively. The amount of NAC bound on the MNPs (mg/mg) was estimated in both NAC-loaded samples. The calculated values are summarized in Table 1, and they agree well with the results from magnetic measurements.

### 2.5. Stability Monitoring

Dynamic light scattering (DLS) was also used to monitor the time and thermal stability of PLLMNPs, NAC-PLLMNPs-1, and NAC-PLLMNPs-3 samples.

The time stability of the samples was evaluated by monitoring their hydrodynamic particle size (*D*_DLS_) and polydispersity index (PDI) in water and the saline solution at two temperatures (*T* = 25 °C and 37 °C). While the PLLMNPs and NAC-PLLMNPs-1 samples were stable for more than 72 h in water at both temperatures, the NAC-PLLMNPs-3 sample started agglomerating after 48 h (Figure 6A). *D*_DLS_ and PDI of NAC-PLLMNPs-3 were found to be 427.05 nm and 0.478, respectively, after 48 h of incubation compared to the results obtained up to 48 h of incubation (*D*_DLS_ was <400 nm and PDI < 0.4). A PDI greater than 0.4 indicates that the sample is polydisperse and large particles/agglomerates are present in the sample [27]. The stability in the saline medium was lower in all studied samples. No increase in *D*_DLS_ was observed in either PLLMNPs or NAC-PLLMNPs-1 samples up to 48 h at any of the monitored temperatures. The NAC-PLLMNPs-3 sample started to agglomerate within one day. In terms of all the measurements carried out in saline medium at 37 °C, the agglomeration began within one day in all samples. The results were confirmed by PDI measurements as well.

The thermal trend measurements measured the thermal stability of all the evaluated samples. One DLS measurement was performed at each temperature. The change in *D*_DLS_ and in PDI over the temperature range is shown in Figure 6B. All samples are stable up to 50 °C as no aggregation was observed throughout the selected temperature range (see inset in Figure 6B). Both *D*_DLS_ and PDI were relatively constant up to a temperature of 55 °C for PLLMNPs and 60 °C for NAC-loaded PLLMNPs. Above this temperature, an increase was observed in *D*_DLS_ and PDI as well. Based on these findings, it can be concluded that NAC adsorbed on the surface of PLLMNPs improved the thermal stability of the samples.

The selected samples stored in the refrigerator were also tested for long-term stability. In all cases, there was no evolution in nanoparticle size during the four weeks of storage. It could be concluded that no aggregation occurs during storage in the refrigerator and the samples show good long-term stability, which is also evidenced by the relatively high value of the zeta potential (see Table 1).

### 2.6. Cytotoxicity Experiments

The toxicity of the samples at concentrations of MNPs 0.1, 10, and 50 μg/mL was measured using an LDH assay kit on the human endothelial kidney (HEK 293) cell line for 24 and 48 h treatment (Figure 7A,B). The viability was determined as the difference of 100% and calculated toxicity. The viability of the two NAC-loaded samples, NAC-PLLMNPs-1 (red) and NAC-PLLMNPs-3 (blue), was comparable to the viability of untreated cells (taken as a control, 100%, white bars). The sample without NAC (PLLMNPs, black bars) showed a decrease in cell viability only at lower concentrations after 48 h of incubation. Otherwise, PLLMNPs were not toxic to the HEK 293 cell line. *N*-acetylcysteine (orange) was also not toxic for cells after 24 or 48 h incubation at all three studied concentrations. Our data suggest that nanoparticles with and without NAC are not cytotoxic up to 50 μg/mL.

### 2.7. MRI Relaxometry

The main goal of NAC adsorption to the PLLMNPs was to characterize the MRI relaxation properties of such NAC-PLLMNPs-1 and acquire preliminary results that can be used further to determine the spatial distribution of the NAC drug in the lung in vivo after the inhalation. In this work, we analyzed and compared the relaxation properties of PLLMNPs with and without NAC adsorbed in vitro to reveal the effect of NAC on the MRI relaxation of carrier nanoparticles. NAC has no significant MRI relaxation properties (unpublished results). As seen in Figure 8, in all three analyzed protocols, the adsorption of NAC to PLLMNPs increases the relaxation time compared to the particles without the drug. However, a more significant change is observed for *T*_1_ relaxation time (Figure 8A), which points to the prevailing longitudinal relaxation of NAC-PLLMNPs-1 sample compared to pure PLLMNPs. The difference is visible in transverse relaxation (Figure 8B,C), although not as dramatic as in *T*_1_. It suggests the mechanical shielding effect of adsorbed NAC to spin–spin interactions, which causes a slightly longer dephasing of transverse magnetization and an increased spin–lattice interaction due to surrounding NAC molecules. These findings indirectly prove the diamagnetism of the NAC drug.

Moreover, the increase in relaxation times after NAC adsorption to the PLLMNPs is also visible to the naked eye, as shown in Figure 9. In all relaxometric protocols, the increase in the relaxation times of the NAC-PLLMNPs-1 sample (Figure 9D–F) compared to pure PLLMNPs (Figure 9A–C) is clearly visible as a hyperintense signal. Especially in PLLMNPs, the concentration gradient of particles from top to bottom, left to right, is unambiguous. It proves a significant shortening of both relaxation times by magnetite nanoparticles as a carrier, although the iron oxides are used primarily as negative (hypointense) contrast agents, with prevailing *T*_2_^(^*^)^ shortening.

However, the unequivocal evidence of the prevailing transverse relaxation mechanism also in our PLLMNPs, without regard to the presence of NAC, is shown in Figure 9D. A higher value of *r*_2_*** is also evident for both cases, which is in accordance with the definition of transverse relaxation and Equation (3). Figure 10A–C show a longitudinal and transverse relaxivity comparison of PLLMNPs and NAC-PLLMNPs-1 particles. In all three cases, relaxivity values decrease with the adsorption of NAC to PLLMNPs (Table 2). However, both in PLLMNPs and in NAC-PLLMNPs-1, the transverse relaxivity values significantly exceed the values of commercially used MRI contrast agents based on iron oxides: Resovist, *r*_1_ = 2.8 mM^−1^ s^−1^, *r*_2_ = 176 mM^−1^ s^−1^ and Feridex, *r*_1_ = 2.3 mM^−1^ s^−1^, *r*_2_ = 105 mM^−1^ s^−1^ measured at 4.7 T [28] and Feraheme, *r*_1_ = 3.1 mM^−1^ s^−1^, *r*_2_ = 68 mM^−1^ s^−1^ measured at 7 T [29]. This is a promising output related to the intended in vivo applications. A key tool in comparing the MRI contrast properties of different compounds is the *r*_2_/*r*_1_ (*r*_2_***/*r*_1_) ratio. As evident in Figure 9E, the *r*_2_/*r*_1_ ratio has the same trend as seen in the relaxivity values, which is expected (lower value for NAC-PLLMNPs-1). Surprisingly, in the case of the *r*_2_***/*r*_1_ ratio, the situation is the opposite (lower value for pure PLLMNPs). The molecular mechanism of this change is currently unclear and requires further studies. However, we showed that the PLLMNPs system, both without and with adsorbed NAC, exhibits excellent MRI contrast properties with significantly prevailing transverse relaxation, which was the main purpose of this study. In addition, it is possible to clearly distinguish between pure PLLMNPs and NAC-loaded PLLMNPs, which can help monitor drug release from magnetic carriers in vitro/in vivo.

## 3. Materials and Methods

### 3.1. Materials

Chemicals—iron (II) sulfate heptahydrate (Merck, Darmstadt, Germany), iron (III) chloride hexahydrate (Sigma-Aldrich, Burghausen, Germany), ammonium hydroxide solution 25% (Slavus, Bratislava, Slovakia), *N*-acetylcysteine injection solution (Sandoz Pharmaceuticals d.d., Ljubljana, Slovenia), poly-l-lysine (Serva, Heidelberg, Germany). NaCl, HCl, and NaOH used to set the pH and ionic strength were analytical grade products of Merck. Milli-Q water was used to prepare all solutions.

### 3.2. Methods

#### 3.2.1. Preparation of Magnetic Nanoparticles

The Fe_3_O_4_ magnetic nanoparticles (with a mean magnetic core diameter of ~10 nm) were prepared by co-precipitating ferrous and ferric ions with an alkaline solution and treating them under hydrothermal conditions. Briefly, 154 mg of FeSO_4_·7H_2_O was dissolved in 2 mL of demineralized water, and then 0.54 mL of 50% solution of FeCl_3_·6H_2_O was added to the ferrous solution, followed by thorough mixing. The mixture was then heated and stirred in a capped glass vessel until 75 °C was reached. Next, 1 mL of NH_4_OH was added, and the black precipitate was formed immediately. The pH was maintained at approximately 11 during the reaction process. After 5 min of mixing, the suspension was washed several times with demineralized water to remove impurities such as chlorides and sulfates. The resulting synthesized suspension of MNPs was used for the following amino functionalization. The magnetite concentration in the prepared MNPs suspension was determined by gravimetric analysis.

#### 3.2.2. Preparation of Amino-Functionalized Magnetic Nanoparticles

The surface modification of MNPs by poly-l-lysine (PLL) was carried out by a simple adsorption method. The defined amount of magnetite suspension and PLL solution in the 1:1 PLL/MNPs (*w*/*w*) ratio was mixed, put in a glass vial, and sonicated (Digital Sonicator^®^ BRANSON Model 450, Branson Ultrasonics, Brookfield, CT, USA) for 5 min in an ice bath under 75% of the power. The samples were prepared in multiple smaller batches, washed, and concentrated by centrifugation at 30,000 rpm for 1 h. The collected precipitates formed the final PLL-modified MNPs sample. The magnetite concentration was estimated by colorimetric determination using the thiocyanate method [30], and PLL quantification on the magnetic nanoparticle surface was performed using the Trypan blue precipitation [31]. The prepared PLLMNPs intended for NAC adsorption consisted of magnetite and PLL with 11 and 1.43 mg/mL concentrations, respectively.

#### 3.2.3. *N*-Acetylcysteine Adsorption on Amino-Modified Magnetic Nanoparticles

Binding of the drug to the MNPs surface was performed by adding NAC to the amino-modified MNPs (PLLMNPs) and subsequent incubation. Briefly, the NAC drug was diluted to a concentration of 10 mg/mL in ultrapure water. Different volumes of this solution were added to 0.5 mL of PLLMNPs aqueous suspension with the MNPs concentration of 2 mg/mL. The samples were subsequently diluted to a volume of 4 mL with demineralized water, resulting in the final MNPs concentration at 0.25 mg/mL, and the NAC concentration ranged from 0.25 to 5 mg/mL (*w*/*w* = 1–20). The final mixtures were stirred at 25 °C for 24 h and then were centrifuged at 40 000 rpm for 1 h to remove free NAC. After centrifugation, the sediments were collected and part of the sediment was subsequently resuspended in demineralized water for further characterization in order to determine the optimal input weight ratio of NAC to MNPs (see Section 2.1). The second part of the washed sample was lyophilized and used for TG and magnetic measurements (see Section 2.3 and Section 2.4) to determine the amount of bound NAC on the PLLMNPs surface.

#### 3.2.4. Physicochemical Characterization

The particle size distribution and the surface charge of the samples were measured via dynamic light scattering (DLS) and laser Doppler electrophoresis using a Zetasizer Nano ZS apparatus (Malvern Instruments, Malvern, UK). The surface charge of pure (naked) nanoparticles (MNPs), PLLMNPs, and NAC-adsorbed PLLMNPs was measured at 25 ± 0.1 °C in a disposable zeta cell (DTS 1060). The instrument’s settings were checked by measuring a standard latex sample with a zeta potential of 42 ± 4.2 mV. In one series of experiments, the effect of various theoretical NAC/MNPs weight ratios on the zeta potential values of the prepared samples was measured. Subsequently, the pH dependence was investigated on samples with various theoretical NAC/MNPs ratios ranging from 0.1 to 16 mg/mg MNPs to obtain the dependence of the IEP value. Finally, the dependence of the IEP on the theoretical NAC/MNPs ratio was obtained. Before measurements, the samples were diluted in 10 mM NaCl and then homogenized in an ultrasonic bath for 10 s followed by 2 min relaxations.

Transmission electron microscope images were acquired on a JEOL 2100F TEM at an accelerating voltage of 200 kV. The images were acquired in a bright field with an Erlangshen CCD camera (1344 × 1036, 12-bit) using Gatan Digital Micrograph software, version 2.02.800.0. The samples were prepared on thin carbon films deposited on top of a copper mesh grid CF200-Cu-UL (Electron Microscopy Sciences, Hatfield, PA, USA).

Atomic force microscope images were obtained using DriveAFM (Nanosurf AG, Liestal, Switzerland) in phase contrast scanning mode on air. Images were acquired using an SNL-10 cantilever at 5 μm × 5 μm scanning size and 1024 × 1024 pix resolution. The size distribution of the PLLMNPs and NAC-PLLMNPs was analyzed using grain analysis in the Gwyddion2.6.5 software. Histogram analysis was performed in Origin 8.5 Pro software.

Magnetic measurements were performed with a Quantum Design MPMS SQUID magnetometer, equipped with a superconducting magnet (a solenoid of superconducting wire), which produces magnetic fields in the range from −7 T to +7 T. The samples in the form of powders were immobilized in an epoxy resin to prevent any movement of the nanoparticles during the measurements.

Thermogravimetric analysis (TGA) was performed to determine the amount of NAC loaded onto PLLMNPs. TG measurements were carried out on dried samples from room temperature up to 900 °C under air with a heating rate of 15 °C/min using the TGDTA Setaram SETSYS 16 apparatus (Setaram, Caluire-et-Cuire, France).

#### 3.2.5. Time and Thermal Stability Studies

The time stability of the selected samples was monitored in demineralized water and saline solution at room temperature (25 °C) and 37 °C by measuring the size of the particles and the polydispersity index in different time intervals. A stock solution (1 mg/mL) of nanoparticles was prepared in demineralized water. Aliquots (0.5 mL) of the stock solution were taken and diluted with 3.5 mL of the study medium. Samples were mixed at temperatures of 25 °C and 37 °C and the hydrodynamic particle size (*D*_DLS_) and the PDI were measured at different time intervals.

Thermal stability was measured by trend measurements, starting at a temperature of 25 °C and increasing to 80 °C in 1 °C steps with an equilibration time of 120 s, while changes in hydrodynamic diameter were recorded. Temperature trends by DLS may help predict the stability of the samples and monitor how the samples respond to thermal stress. Three DLS measurements were performed at each temperature.

#### 3.2.6. Cytotoxicity Experiments

The toxicity of selected samples with adsorbed NAC, PLLMNPs without NAC, and *N*-acetylcysteine (NAC) alone was determined using an LDH assay kit (CyQUANT™, Invitrogen™, Thermo Fisher Scientific, Waltham, MA, USA) according to the manufacturer’s protocol. Shortly, HEK 293 cells were seeded in a 96-well plate at 10,000 cells/well density and incubated in a humid atmosphere (37 °C, 5% CO_2_) overnight. Samples at three concentrations (0.1, 10, and 50 μg/mL) were added to cells in triplicate and incubated for 24 h or 48 h. After incubation, 50 μL of media over the cells was mixed with the LDH reaction mixture and incubated for 30 min. The reaction was stopped using an LDH stop solution. The LDH activity was calculated as a difference between the 490 and 680 nm absorbance values (background signal from the instrument, Synergy Mx plate reader (BioTek, Winooski, VT, USA)). Cytotoxicity was calculated using the following formula:(1)$$\%\;cytotoxicity=\frac{Compound-treated\;LDH\;activity-Spontaneous\;LDH\;activity}{Maximum\;LDH\;activity-Spontaneous\;LDH\;activity}\times 100$$

The overall viability was determined as the difference between 100 and the calculated % of cytotoxicity. The data are presented as the average of a triplicate measurement with standard deviation.

#### 3.2.7. Magnetic Resonance Imaging

Three different-weighted MRI protocols determined the relaxation time (*T*_1_, *T*_2_, *T*_2_***) and particle’s relaxivity (*r*_1_, *r*_2_, *r*_2_***) with and without adsorbed NAC:*T*_1_ mapping—Rapid acquisition with refocused echo (RARE) pulse sequence, with repetition time TR = 5500, 3000, 1500, 800, 400, and 200 ms, and echo time TE = 7 ms.*T*_2_ mapping—Multi-slice multi-echo (MSME) pulse sequence, with repetition time TR = 2000 ms, starting echo time TE = 8 ms, spacing = 8 ms, and 25 images.*T*_2_* mapping—Multi-gradient echo (MGE) pulse sequence, with a repetition time TR = 1200 ms, starting echo time TE = 5.1 ms, spacing = 5 ms, and 10 images.

Images were acquired using the following parameters: flip angle (FA) = 90°, FOV = 8 × 6 cm, matrix = 256 × 192 pixels, and slice thickness = 1 mm. We used the concentration gradient of MNPs (0, 5, 10, 15, 20, 25, 30, 35 µg/mL) to determine the following MRI parameters.

Relaxation times (*T*_1_, *T*_2_, *T*_2_***) were determined by fitting the following functions from signal intensity values of different times:(2)$$M_{z}=M_{0}(1-e^{-t/T_{1}})$$
(3)$$M_{xy}=M_{0}e^{-t/T_{2}}$$
where *M_z_* is the z-component of the magnetization vector *M* (longitudinal magnetization), *M_xy_* is the transverse magnetization after the pulse, *M*_0_ is the initial maximum value of magnetization *M*, *t* is the time, *T*_1_ is the longitudinal relaxation time, and *T*_2_ is the transverse relaxation time. *T*_2_*** relaxation time is a combination of the “true” transverse relaxation time *T*_2_ and additional relaxation (*T*_2_′) caused by magnetic inhomogeneities:(4)$$\frac{1}{T_{2}^{*}}=\frac{1}{T_{2}}+\frac{1}{T_{2}^{\prime }}$$

Relaxivity value *r* (mM^−1^ s^−1^) defines the MRI contrast efficiency of MNPs in MRI and is calculated from the slope of inversion relaxation time (defined as relaxation rate *R* (s^−1^)) depending on iron concentration (mM):(5)$$R=R_{0}+rC$$
where *R*_0_ is the relaxation rate in the absence of MNPs, *R* is the relaxation rate in the presence of MNPs, and *C* is the iron concentration in the solution of MNPs. 

For MRI data processing, analysis, and visualization, we used a Paravision “Image Sequence Analysis” tool (Bruker, Billerica, MA, USA) and a MATLAB R2022b software tool (Mathworks Inc., Natick, MA, USA).

## 4. Conclusions

In this work, poly-l-lysine-coated magnetic nanoparticles (PLLMNPs) were successfully synthesized and used for the adsorption of the drug *N*-acetylcysteine. The influences of various theoretical NAC/MNPs weight ratios on particle size, polydispersity, zeta potential, and IEP were systematically assessed. Based on optimization experiments, it was found that the NAC-PLLMNPs-1 sample with the following parameters as *D*_DLS_ = 154 nm, PDI = 0.22, zeta potential = +32 mV, and the experimentally obtained ratio of ~0.01 mg NAC/mg MNPs could be the most suitable for further MRI experiments due to the good time and temperature stability in physiological solution. PLLMNPs and NAC-adsorbed PLLMNPs exhibit excellent MRI contrast properties with significantly prevailing transverse relaxation. Moreover, it is possible to distinguish between magnetic nanoparticles without and with adsorbed NAC. This fact can help in monitoring drug release in vitro/in vivo. The achieved results are the starting point for further investigation, in which we focus on studying the spatial distribution of NAC after inhalation using MRI relaxometry. In summary, this prepared NAC-PLLMNPs, combined with MRI, represents a useful tool that provides significant advances in imaging drug distribution in the lung and in acute respiratory distress syndrome in further studies.

## Figures and Tables

**Figure 1 ijms-24-11414-f001:**
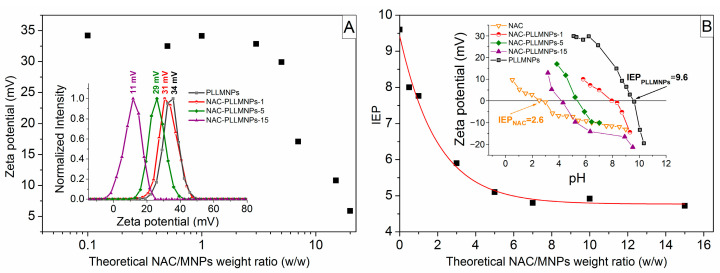
(**A**) The effect of various theoretical NAC/MNPs weight ratios on zeta potential. Inset: the zeta potential distributions of the selected samples with apparent effect of NAC loading on zeta potential; (**B**) the effect of various theoretical NAC/MNPs weight ratios on the IEP. Inset: pH dependence of zeta potential of the selected samples.

**Figure 2 ijms-24-11414-f002:**
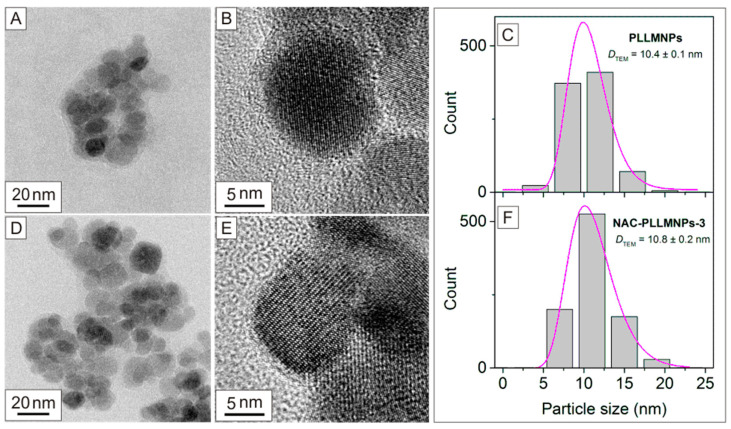
TEM, HRTEM images, and particle size distributions of PLLMNPs (**A**–**C**) and NAC-PLLMNP-3 (**D**–**F**). The size distributions (grey bars) were fitted with a log-normal function (purple curves) (**C**,**F**).

**Figure 3 ijms-24-11414-f003:**
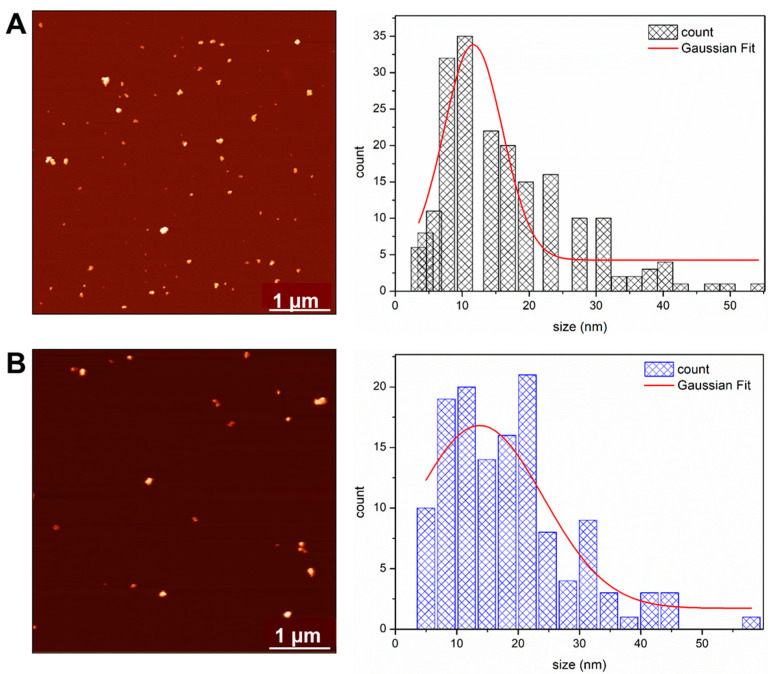
AFM images and particle size distributions of PLLMNPs (**A**) and NAC-PLLMNP-3 (**B**). The size distributions were fitted with a Gaussian function (solid red line).

**Figure 4 ijms-24-11414-f004:**
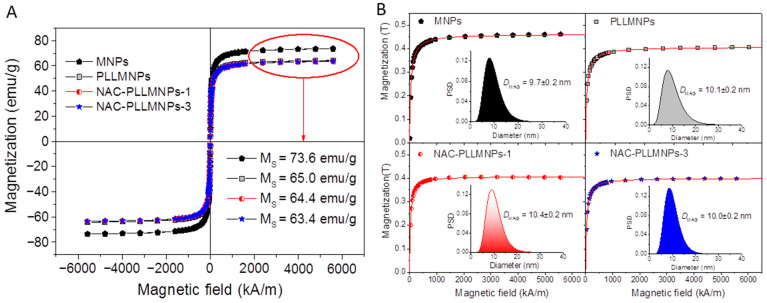
(**A**) Hysteresis loops measured at room temperature and (**B**) magnetization curves for all the samples fitted by the Langevin function. The inset displays magnetic core diameter distributions with calculated mean *D*_MAG_.

**Figure 5 ijms-24-11414-f005:**
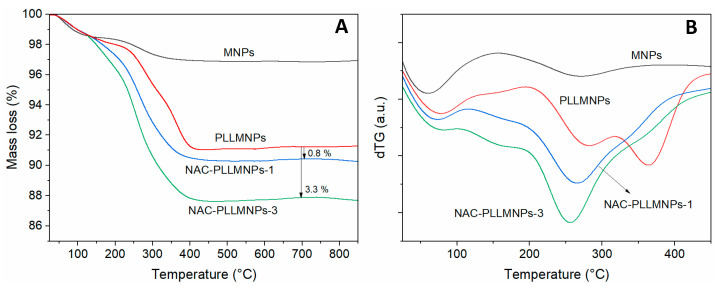
Thermograms (TG (**A**) and dTG (**B**)) for bare MNPs, PLL-modified MNPs, and NAC-loaded PLLMNPs samples with the input *w*/*w* ratio 1 and 3.

**Figure 6 ijms-24-11414-f006:**
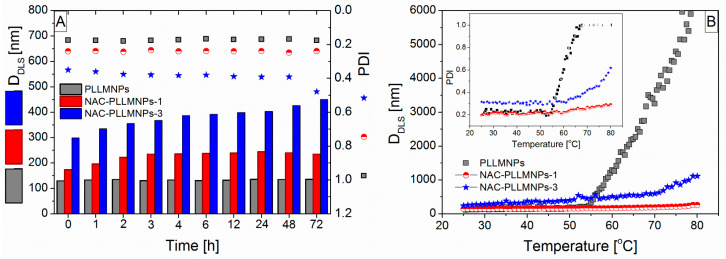
(**A**) Change in mean peak size and polydispersity of PLLMNPs, NAC-PLLMNPs-1, and NAC-PLLMNPs-3 with time in the water at *T* = 25 °C; (**B**) thermal trend measured showing an increased aggregation temperature for PLLMNPs in demineralized water with the different NAC addition.

**Figure 7 ijms-24-11414-f007:**
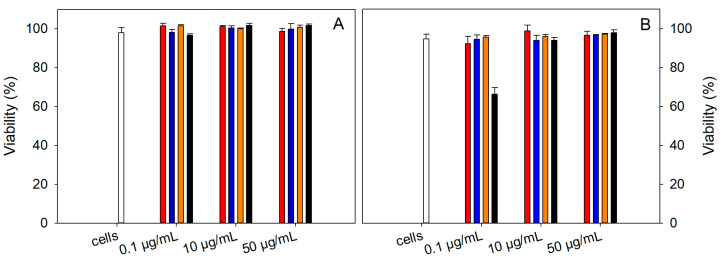
The viability of human endothelial kidney cell line (HEK 293) in the presence of 0.1, 10, and 50 μg/mL MNPs concentrations in NAC-PLLMNPs-1 (*w*/*w*) (red), NAC-PLLMNPs-3 (*w*/*w*) (blue), NAC alone (orange), and PLLMNPs (black) after 24 (**A**) and 48 h (**B**) treatment. Viability was determined using an LDH assay. The results represent the average of the three independent measurements with standard deviations.

**Figure 8 ijms-24-11414-f008:**
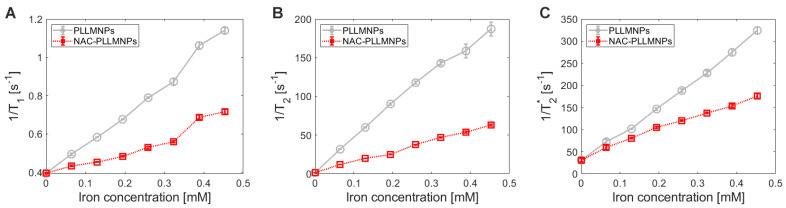
The inverse relaxation time values of PLLMNPs with (red) and without adsorbed NAC (grey): (**A**) the inverse longitudinal relaxation time *T*_1_, (**B**) the inverse transverse relaxation time *T*_2_, (**C**) the inverse transverse relaxation time *T*_2_***.

**Figure 9 ijms-24-11414-f009:**
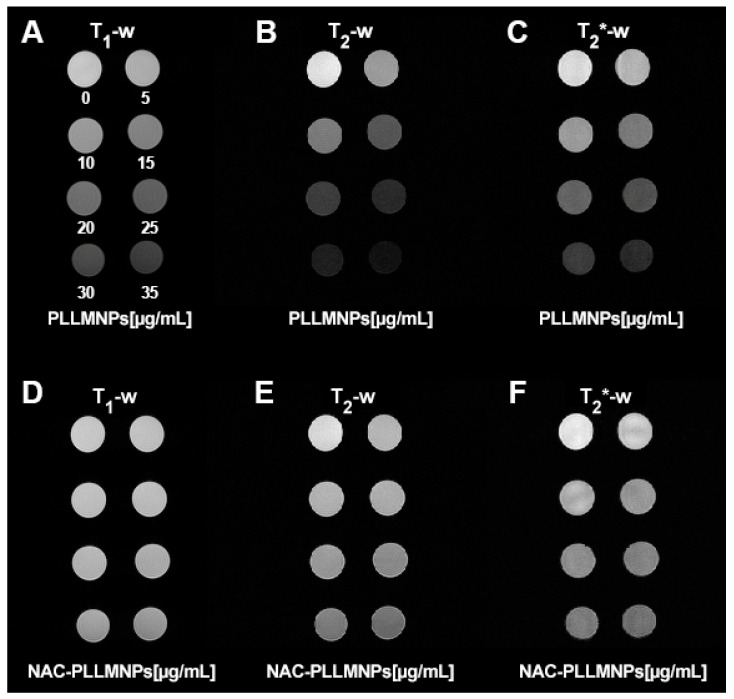
The signal intensity of concentration gradient of PLLMNPs (upper part) and PLLMNPs with NAC (lower part) acquired by different relaxometric protocols: (**A**) *T*_1_-weighted RARE protocol—PLLMNPs; (**B**) *T*_2_-weighted MSME protocol—PLLMNPs; (**C**) *T*_2_***-weighted MGE protocol—PLLMNPs; (**D**) *T*_1_-weighted RARE protocol—NAC-PLLMNPs-1; (**E**) *T*_2_-weighted MSME protocol—NAC-PLLMNPs-1; (**F**) *T*_2_*-weighted MGE protocol—NAC-PLLMNPs-1.

**Figure 10 ijms-24-11414-f010:**
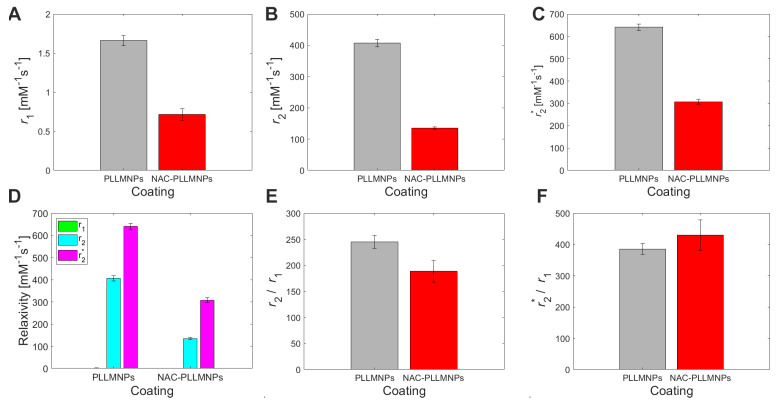
Relaxivity of the concentration gradient of PLLMNPs without (grey bars) and with adsorbed NAC (red bars) acquired by different relaxometric protocols: (**A**) *r*_1_ acquired by *T*_1_-weighted RARE protocol, (**B**) *r*_2_ acquired by *T*_2_-weighted MSME protocol, (**C**) *r*_2_*** acquired by *T*_2_***-weighted MGE protocol, (**D**) comparison of PLLMNPs and NAC-PLLMNPs-1 relaxivities, (**E**) *r*_2_/*r*_1_ ratio of PLLMNPs (grey bars) and NAC-PLLMNPs-1 (red bars), (**F**) *r*_2_***/*r*_1_ ratio of PLLMNPs and NAC-PLLMNPs-1.

**Table 1 ijms-24-11414-t001:** Physicochemical properties of PLLMNPs and NAC-PLLMNPs obtained by different techniques.

Sample	Initial Input NAC/MNPs	Experimental Output NAC/MNPs		*D* _MAG_	*D* _TEM_	*D* _DLS_	PDI	Zeta Potential
		SQUID	TGA					
	(mg/mg)	(mg/mg)	(mg/mg)	(nm)	(nm)	(nm)		(mV)
PLLMNPs	0	0	0	10.1 ± 0.2	10.4 ± 0.1	112.2 ± 0.9	0.18	34 ± 1.6
NAC-PLLMNPs-1	1	0.010	0.009	10.4 ± 0.2	10.8 ± 0.2	154.1 ± 1.1	0.22	32 ± 1.8
NAC-PLLMNPs-3	3	0.028	0.035	10.0 ± 0.2	10.7 ± 0.2	238.0 ± 22.1	0.31	31 ± 1.3

Initial Input NAC/MNPs (mg/mg)—initial NAC to MNPs input ratio in the solution; experimental output SQUID (mg/mg)—NAC adsorbed on MNPs (mg/mg) determined from SQUID measurement; experimental output TGA (mg/mg)—NAC adsorbed on MNPs (mg/mg) determined from thermogravimetric analysis; *D*_MAG_ (nm)—magnetic core diameter obtained from magnetic measurements; *D*_TEM_ (nm)—the mean size of magnetic particle core evaluated from TEM images; *D*_DLS_ (nm)—Z-average of magnetic particle size measured by DLS measurements; PDI—polydispersity index obtained by DLS measurements; Zeta Potential (mV)—zeta potential of the samples.

**Table 2 ijms-24-11414-t002:** Relaxivity comparison of PLLMNPs without and with adsorbed NAC.

	*r*_1_ (mM^−1^ s^−1^)	*r*_2_ (mM^−1^ s^−1^)	*r*_2_* (mM^−1^ s^−1^)	*r*_2_/*r*_1_	*r*_2_*/*r*_1_	B_0_ (T)
PLLMNPs	1.66 ± 0.07	406.80 ± 12.02	640.60 ± 14.64	245.06	385.90	7
NAC-PLLMNPs-1	0.71 ± 0.08	134.80 ± 3.69	307.07 ± 11.16	189.86	432.49	7

## Data Availability

The data that support the findings of this study are available on request from the corresponding author.

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
