# Peer review of "N-Acetylcysteine-Loaded Magnetic Nanoparticles for Magnetic Resonance Imaging"

_ijms, 2023, doi:10.3390/ijms241411414_

Round 1

Reviewer 1 Report (Previous Reviewer 1)

The paper is acceptable and can be published.

See comments

Author Response

Reviewer 2 Report (New Reviewer)

The work of M.Kubovcikova et all. reports about synthesize of ferrum oxide nanoparticle, modified by poly-L-lysine which adsorb N-acetylcysteine. There are morphology and physicochemical characterizations of obtained nanoparticles. Unfortunatelly, I can’t recommend this article as it stands for publishing. Authors are advised to pay attentions to the followings:

1) There is no zeta potential of iron oxide nanoparticles in the article. It is not clear, why poly-L-lysyne was chosen.

2) It is well known, that zeta potential of PLL coatings dependence on pH of solution. As I understand, addition different amount of NAC lead to two opposing factors: increasing zeta potential by acidification and decreasing zeta potencial by adsorption of NAC. I think it is nessesary to work in buffer solution. 

3) What is the driving force of adsorption of NAC?  

4) Fig 8 should be present as dependence of 1/T1 or1/T2 (s-1) vs concentration of Fe (mM)

5) As I know the transverse relaxation time of pure water is about 2400-2700 ms. It is not clear, why there is 650 ms at T2 mode in figure 8?

6) In the experimental part, it is written that the content of N-acetylcysteine was determined by various methods. There is no further information about its definition. It is remarkable that the particles have been deposited. However, has the supernatant been examined for the presence and content of N-acetylcysteine? (432-434).

7) In my mind, the reason of increasing r2/r1 ration is simple aggregation of nanoparticle. It is known, that Agglomeration of iron oxide nanoparticles lead to higher inhomogeneity of local magnetic field and therefore greater r2 relaxivity. r2 is predominant and reduces the r1.

Round 2

Reviewer 2 Report (New Reviewer)

it is all clearly. The article is ready to be accepted.

This manuscript is a resubmission of an earlier submission. The following is a list of the peer review reports and author responses from that submission.

Round 1

Reviewer 1 Report

In the present work, the authors tried to investigate drug delivery in the lung. Some points should be clarified:

1.      Please revise the grammar of the paper carefully according to the journal standard.

2.      The literature review is not well written. The number of newly published works that have been considered is so few. So, the novelty is in doubt.  

3.      There are many abbreviations in the text without a proper introduction. It is necessary to introduce the abbreviation.

4.      On page 4, the authors said that the NAC/MNPs (w/w) = 5 does not have good stability. What is the reason for this?

5.      The authors emphasize that they studied the effect of magnetic nanoparticles on imaging. But the physic of the magnetic section and the mathematical equations are not explained at all. A discussion about this matter is necessary.

6.      More explanation about the nanoparticles is needed!

1.      Please revise the grammar of the paper carefully according to the journal standard.

Reviewer 2 Report

Very poor work. Can not be consider for publication. 

Reviewer 3 Report

Sorry, but I can not recommend this work for publication.

The main problem is that you did not prove the title! Where is the direct proof of conjugation? I do not want to mention the lack of novelty.

Can you please answer the question what is the size of your material? From TEM = ca.10 nm (do not change after NAC addition), AFM ca. 20 nm, while DLS = up to 238 nm; such differences must be explained! There is no discussion at all.

Dear Authors adsorption is not the same as conjugation!!! That is why your results are so spread-out. But it means that your biological work was for nothing, sorry.

For future:

-      Fig 1A inset, (i) curves are too beauty; what do you mean normalized? Normalized to what? Because not to unity!

-      The highest value of the NAC adsorption efficiency, calculated as a ratio of the actual adsorbed amount of NAC to theoretical loading NAC amount expressed in %, was found to be 89% in the sample with NAC/MNPs (w/w) ratio equal to 5 is IMPOSSIBLE, you should recalculate this; BTW, adsorption isotherms are usually shown as a function of equilibrated concentration, it is better.

-      Why do you show the calibration curve up to 0.1 mg/mL while your working concentrations are up to 5 mg/mL!!! more, is it a surprise that NAC follows L-B low?

-      The results presented in Fig 3 should be explained deeper; I do not agree that “the NAC-PLLMNPs-3 sample started agglomerating

-      Why there is no explanation of Fig 3B?

-      Why you used HEK 293 while (i) you talk about lungs? (ii) it is known to be a quite resistive line?